# The architecture of assisted colonisation in sea turtles: building new populations in a biodiversity crisis

Anna Barbanti[1], Janice M. Blumenthal[2], Annette C. Broderick[3], Brendan J. Godley[3], Alejandro Prat-Varela [2], Maria Turmo[1], Marta Pascual [1,4] & Carlos Carreras [1,4 ✉]

Due to changing environmental conditions, many species will have to migrate or occupy new suitable areas to avoid potential extinction in the current biodiversity crisis. Long-lived animals are especially vulnerable and ex-situ conservation actions can provide solutions through assisted colonisations. However, there is little empirical evidence on the process of founding new populations for such species or the feasibility of assisted colonisations as a viable conservation measure. Here, we combined genetics with reproductive data to study the rise of two wild populations of green turtles (*Chelonia mydas*) in the Cayman Islands as a possible outcome of a reintroduction program started 50 years ago. We show that both populations are highly related to the captive population but rapidly diverged due to genetic drift. Individuals from the reintroduced populations showed high levels of nest fidelity, within and across nesting seasons, indicating that philopatry may help reinforce the success of new populations. Additionally, we show that reintroduction from captive populations has not undermined the reproductive fitness of first generation individuals. Sea turtle reintroduction programs can, therefore, establish new populations but require scientific evaluation of costs and benefits and should be monitored over time to ensure viability in the long-term.

[1] Department of Genetics, Microbiology and Statistics and IRBio, Universitat de Barcelona, Av. Diagonal 643, 08028 Barcelona, Spain. [2] Department of Environment, PO Box 10202 Grand Cayman KY1-1002, Cayman Islands. [3] Centre for Ecology and Conservation, University of Exeter, Penryn Campus, Penryn TR10 9FE, UK. [4] These authors jointly supervised this work: Marta Pascual, Carlos Carreras. ✉email: carreras@ub.edu

        1

The rate of biodiversity loss has accelerated during the last decade[1]. Anthropogenic impacts such as global warming, habitat alteration, and human-mediated dispersal of alien invasive species are some of the main causes of the biodiversity crisis at a global scale[2]. Ecosystems are being dramatically altered to the point that they are no longer suitable for some organisms[3]. Consequently, species must either rapidly adapt or move to new suitable habitats to avoid extinction and for this reason, some species are changing their distributions by founding new populations[4]. However, species with limited potential to adjust their distributions to the new climatic conditions are potentially more vulnerable and thus more prone to extinction[5,6]. While adaptation is difficult to predict, range expansions can be detected and even facilitated through reintroductions from ex-situ conservation programs[7,8]. Unfortunately, reintroductions from captive breeding programs are rarely evaluated to assess their longer-term success[9–11]. This evaluation is crucial, as reintroduced individuals can display reduced reproductive success and newly founded populations can suffer reductions of genetic variability due to the founder effect. The foundation process of new populations was theoretically described in the last century[12], however, few studies have provided empirical data and most of them have focused on short-lived organisms[13].

The study of founding processes in long-lived vertebrates is challenging, but essential in the current era of global biodiversity decline, as these species are potentially highly vulnerable to habitat alterations, may have slow responses to environmental change and are potential targets for assisted colonisation programs[5,14]. As reptiles, sea turtles are strongly influenced by temperature[15,16], and have Temperature-dependent Sex Determination (TSD) with rising temperatures causing the feminization of nesting populations[16]. Modelling studies have predicted a potential collapse of some existing nesting populations due to environmental change, and at the same time new areas would become potentially suitable for nesting[17–19]. While sea turtles are highly migratory species[20], their potential to colonise new nesting areas is thought to be limited due to their philopatric behaviour[21,22]. To date, few cases of natural changes in the distribution of sea turtle nesting areas have been detected[19] and for this reason, assisted colonisation has been proposed as a conservation tool to conserve populations threatened by anthropogenic activities or to reinforce natural expansion processes[5,14,23].

The Cayman Islands green turtle (*Chelonia mydas*) reintroduction program offers a unique opportunity to study the process and consequences of assisted colonisations in sea turtles. The local green turtle nesting population was considered nearly extinct[24] but the number of nesting females has increased exponentially in Grand Cayman over the past 20 years[25], partially as a result of the reintroduction program initiated in 1983 from the Cayman Turtle Farm (CTF) (now the Cayman Turtle Conservation and Education Centre Ltd.)[26,27]. This reintroduction was based on releasing captive-bred green turtles from the island of Grand Cayman where the CTF is based (Fig. 1A), often after a head-starting period (i.e., the rearing of offspring in captivity for approximately 1 year)[28]. This strategy is thought to increase recruitment and, because of philopatry, released animals would likely one day come back as adults to breed where they had been incubated. Since philopatry increases genetic differentiation of geographically distant populations, this strategy may tend to cause the genetic isolation of the reintroduced population from others surrounding it. The process and rate of differentiation into genetically separated nesting grounds has not yet been observed in a newly founded sea turtle population. Moreover, understanding the dynamics of assisted colonisation in establishing new populations could provide

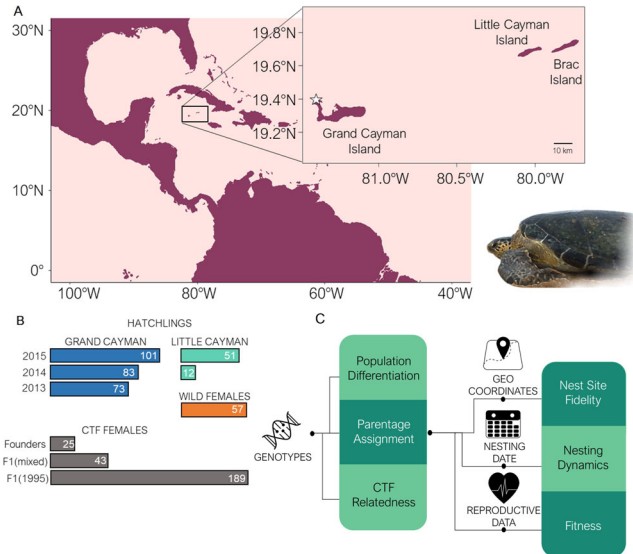

**Fig. 1 Study design. A** Location of the Cayman Islands in the Caribbean Sea. The white star shows the location of the Cayman Turtle Farm (CTF). **B** Samples used for the study. In dark blue, number of hatchlings analysed from Grand Cayman (2013-2015) and in light blue, number of hatchlings analysed from Little Cayman (2014-2015). In orange, wild adult females sampled from nesting beaches in Grand Cayman, and in grey, CTF female breeders, including original founders of the captive population, a multiannual (1986-2002) cohort of first-generation breeders (F1(mixed)) and the cohort of 1995 first generation breeders (F1(1995)). Wild and captive female genotypes are from a previous study[27]. **C** Flowchart of analyses performed in our study. Each hatchling was collected from a different nest and genotyped at 13 microsatellite loci. Genotypes of hatchlings were combined with those of wild female to perform parentage assignment and with CTF female breeder genotypes to assess relatedness to the CTF. Geographic coordinates, nesting dates and reproductive data for each nest assigned to a wild female were extracted from the Department of Environment (DOE, Cayman Islands Government) database to assess nest-site fidelity, nesting dynamics and fitness. Lines indicate what sources of data (genotypes, geographic coordinates, nesting dates and reproductive data) and outputs of the different analyses (parentage assignment and CTF relatedness) interact to provide the different results (coloured boxes), as detailed in the Methods section. Photograph by Anna Barbanti.

significant insights to inform management responses to critical environmental changes. The long-term CTF reintroduction program provides a unique opportunity to address these questions by using a multidisciplinary approach. For these reasons, we studied the possible impact of the CTF on the wild nesting areas of Grand Cayman and the nearby Little Cayman (108.4 km distant), the foundation and differentiation process of these populations, including the role of philopatry, and any potential effect of the reintroduction on individual fitness.

In this work, we examine two wild populations of green turtles in the Cayman Islands following a reintroduction program started 50 years ago. Our results show both populations are highly related to the captive population, but diverged due to genetic drift, and that philopatry may reinforce the success of new populations. Additionally, we show that reintroduction from captive populations has not undermined the reproductive fitness of first-generation individuals. Sea turtle reintroduction programs can, therefore, establish new populations but require scientific evaluation of costs and benefits and should be monitored over time to ensure viability in the long term.

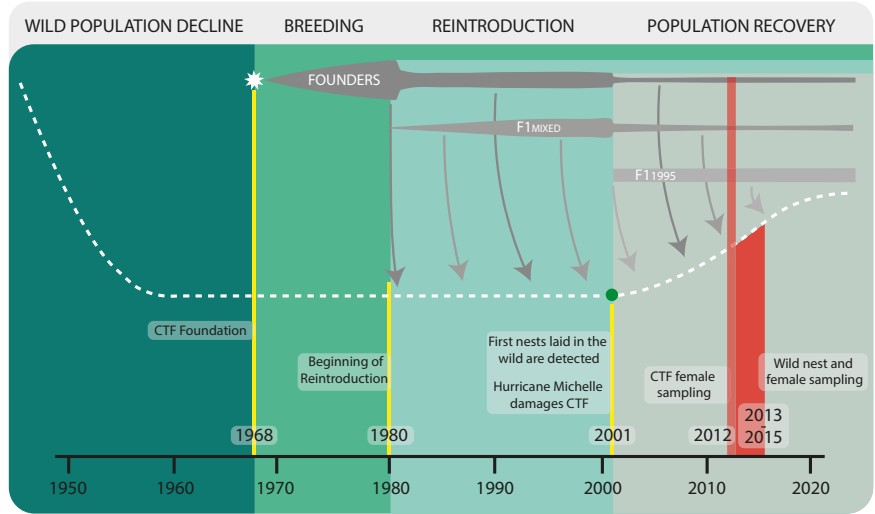

**Fig. 2 Timeline of the Cayman Islands green turtle populations and the reintroduction program.** The key events of the Cayman Islands green turtle populations are detailed, including their decline, the captive breeding, the reintroduction program and the recent recovery. The dashed white line represents the wild population showing the initial decline until its virtual extinction[24] and later increase[25]. The green dot on the dashed line represents the discovery of the first nests laid by wild females. The grey bars at the top represent the three groups of captive breeders in the Cayman Turtle Farm (CTF): founders, F1(mixed) breeding stock and F1(1995) breeding stock. Founders were incorporated to the CTF since its beginning (white star) and suffered two major reductions, the first in the 80 s following a change of farm strategy and the second due to hurricane Michelle[27]. The F1(mixed) individuals were incorporated since the 80s and also suffered the consequences of hurricane Michelle[27] while the F1(1995) individuals were incorporated to the breeding stock to replace the losses of the Hurricane. Grey arrows represent the offspring from the breeding stock released into the wild. The light red vertical bar represents the sampling of captive females for genetic analyses[26] and the red bar represents sampling of wild females[26] and the three seasons of hatchling sampling for the present study (Fig. 1B).

## Results and discussion

**Population diversification from the captive population.** Here we present genetic and ecological data from three to four generations of green turtles collected across three different stages of the assisted colonisation process (CTF Females, Wild Females, and Hatchlings, Fig. 1B–C, Fig. 2). Firstly, we wanted to assess what portion of the new wild generation of green turtles was related to the CTF, in so doing illuminating the contribution of the assisted colonisation to both Grand Cayman and Little Cayman. Running maternity analysis, we were able to assign clutches to their respective mother within our sampled wild females ($n = 149$) and to infer potential unsampled mothers for the remaining unassigned clutches ($n = 171$) (Fig. 3A). By identifying these mother-offspring pairs, we found that progeny from 43% of clutches was related to the turtles in the CTF. Although these results already represent a significant portion of the wild population, a relatedness analysis between wild hatchlings and CTF breeders increased the percentage of related progeny to 88.1%, for an overall total of 282 different clutches related to the CTF. The remaining individuals may be unrelated to the farm or related to CTF breeders that were not assessed genetically because they escaped or were deceased. Overall, we found 79.4% of Little Cayman clutches and 90.3% of Grand Cayman clutches were related to the adults in the CTF (Fig. 3B), with no significant difference in the proportion between the two islands ($\chi^2 = 0.259$, $p$ value = 0.610). These results confirm that the nesting populations of these two islands are mainly the result of an assisted colonisation through individuals from the captive breeding program.

Although maternity and sibship analysis allow us to reconstruct the family pedigree, the measure of genetic difference between nesting groups reflects the evolutionary differentiation process of the new populations. The observation of this process in its very first phases is crucial, not only to set the baseline for future assisted colonisation projects, but also because its detection

in long-lived, migratory vertebrates is rare. In the case of the Cayman Islands, despite the high degree of relatedness of both nesting populations to the CTF, significant genetic differences were found among the three groups, especially with biparentally inherited markers (supplementary information: Table S1). This was also observed with the limited overlap of the three groups, in particular when comparing Little Cayman clutches with CTF female breeders (Fig. 3C, supplementary information: Table S1). This result is consistent with the greater geographic distance between Little Cayman and the release point of captive individuals, and the lower level of relatedness with the CTF (Fig. 3B), which could be the result of the contribution of remnant individuals from the original Little Cayman population. Genetic differentiation between hatchlings from Grand and Little Cayman was statistically significant at the nuclear level and was also seen as shifts in mitochondrial D-loop haplotype frequencies (Fig. 3D), meaning that they should be considered to be two different populations. Most of the haplotypes found in the two islands belong to the Caribbean and South Atlantic lineages[29] and were present in the CTF captive population at different frequencies. The degree of differentiation found between these three populations in only a very few generations suggests that genetic drift during the founder process has been a strong evolutionary force on our populations, able to drive genetic differentiation over small temporal and spatial scales. Selection is unlikely to have played a major role in this differentiation process, as the two islands are separated by only 108 km and have similar environmental conditions. Only strong and opposite selection coefficients could explain such differentiation in a few generations. Previous studies have suggested that philopatry is one of the main drivers of the deep genetic structuring found in sea turtles, as this behaviour reduces gene flow significantly among populations. Therefore, mutation, selection and genetic drift across many generations would generate differentiation among isolated sea turtle populations over extended evolutionary

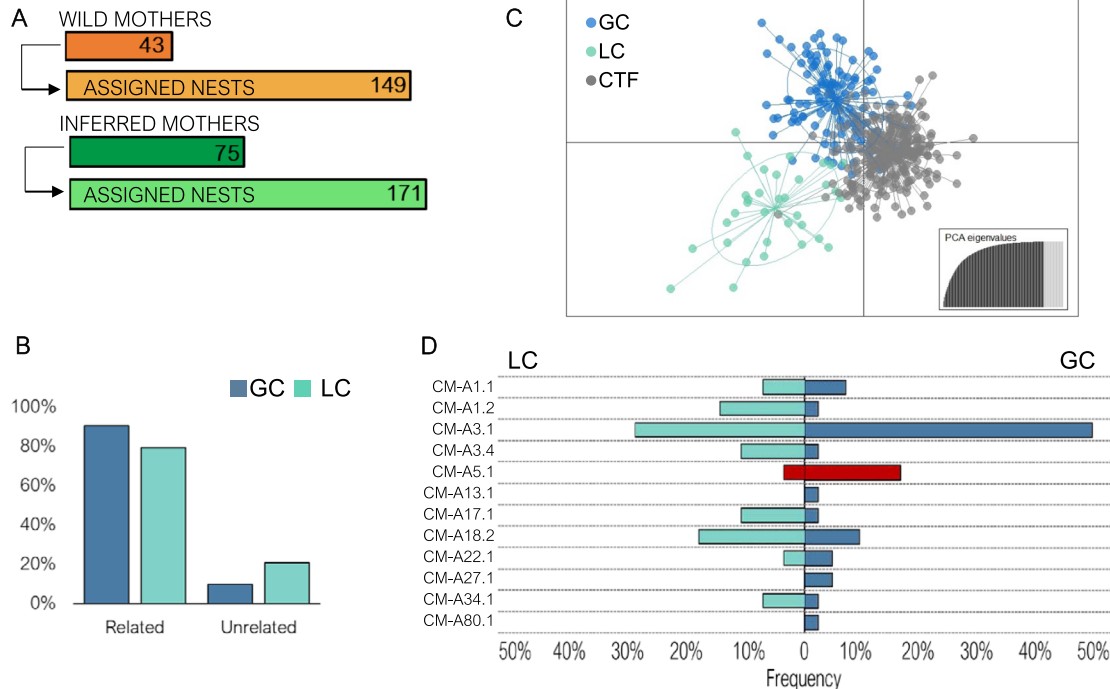

**Fig. 3 Population differentiation from the captive population. A** Number of wild (orange) and inferred (green) mothers and number of nests assigned respectively according to parentage analysis using COLONY. **B** Percentage of nests in Grand Cayman (GC) and Little Cayman (LC) found to be related or unrelated to the Cayman Turtle Farm (CTF) breeding stock as shown by Coancestry and ML-Relate. Both new populations show a high degree of relatedness to the CTF captive population. **C** Discriminant Analysis of Principal Components ($n = 406$ individuals) of GC and LC nests, and CTF breeders, showing the dispersion of the genotypes of the new populations due to a founder effect; the inset shows Principal Component Analysis eigenvalues, retained axes are in black. **D** Frequency of D-Loop haplotypes found in GC and LC nests indicating variations in the frequencies, probably caused by the founder effect. Blue shaded haplotypes belong to the Northern lineage and the red haplotype to the Southern lineage[29].

scales[30]. Here, we demonstrate that genetic drift during founding processes can have also an important role in generating significant genetic structuring in sea turtles in only one generation after the foundation of new populations. After this initial potential differentiation by genetic drift during the founder process, philopatry is expected to increase this differentiation in the future by maintaining the isolation of populations across generations.

**The role of nest-site fidelity in founding new populations.** Philopatry limits the colonising potential of specific organisms but ensures reutilisation of suitable habitats, reinforcing population growth[31]. Consequently, if a new population is established, philopatry will accelerate its growth in the ensuing generations[19]. The CTF reintroduction program was based on the premise that the released animals would be philopatric to the new areas, as shown for other species[23], and that the individuals of the new population would maintain this successful behaviour that is common in all extant turtles. We analysed the breeding dispersal (i.e. displacements between different breeding episodes[32]) on both islands to assess the degree of nest-site fidelity by combining parentage analyses and nesting information. This analysis aimed to understand the role of nest-site fidelity in the differentiation of philopatric species within the wider context of an assisted colonisation. To do so we used nest geographic coordinates to calculate the distances between nests laid by the same female in the same and different nesting seasons (see Methods).

Wild females concentrated their nesting activity in certain areas (Fig. 4A) and the majority of females exhibited a high degree of nest-site fidelity within and between nesting seasons. For all our metrics (i.e. mean distance between nests within a season, the distance between the two most distant nests within a season, and mean distance between nests in different seasons), more than 60% of observations occurred within less than 1 km of each other (Fig. 4B, C). These results show that females in the Cayman Islands have a high degree of nest site fidelity despite coming from a reintroduction program. Only one wild female was found nesting on beaches of both Little Cayman and Grand Cayman, covering a minimum distance of 142.3 km when moving between islands, showing strong nest-site fidelity when nesting within Little Cayman (mean distance between nests = 484 m) while a single nest was laid in Grand Cayman.

Females nesting on the two islands could reflect an actual failure in finding the natal beach or be a consequence of external disturbance during nesting. Nonetheless, long-distance nesting (either on the same or different islands) could also be an evolutionary strategy that maintains gene flow and avoids collapse due to extreme philopatry. Recent research on within-season nest-site fidelity using genetic and satellite tracking highlights that long-distance nesting appears to be more common than previously described[33,34]. Furthermore, we did not find any significant impact of female heterozygosity ($n = 27$, $T = -1.082$, $p$ value $= 0.289$) or CTF relatedness ($n = 27$, Kruskal-Wallis chi-square $= 1.446$, $p$ value $= 0.694$) on within-season mean distance, or on maximum distance ($n = 27$, $T = -0.970$, $p$ value $= 0.341$, Kruskal-Wallis chi-square $= 1.403$, $p$ value $= 0.704$). These results suggest that the reintroduction program might not influence nesting distance and that long-distance nesting events are more likely the result of stochastic processes.

**Long-term effects of the reintroduction.** In recent years, the sex ratio of sea turtle populations has become a cause of concern due to climate change[15,16,35,36]. Sea turtles, as with many other reptile species, exhibit Temperature-dependent Sex Determination with

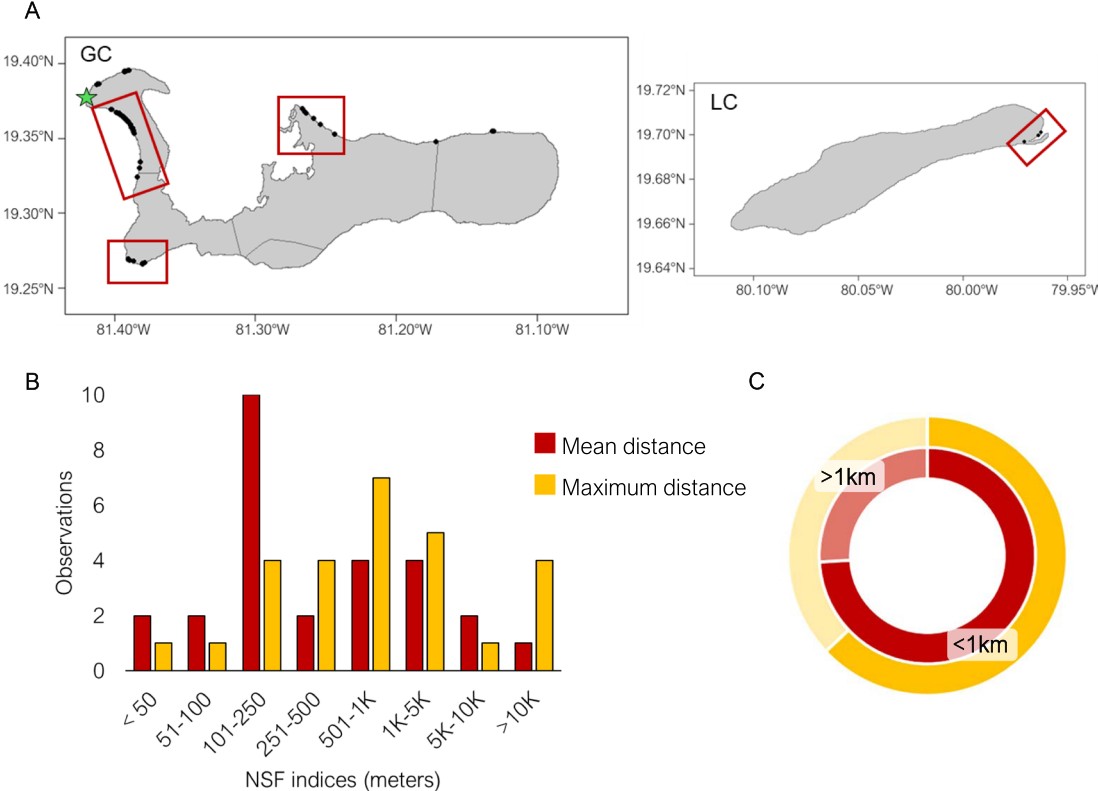

**Fig. 4 Nest-site fidelity. A** Map detailing the geographic locations of the 149 nests laid by the 43 wild females (black dots) and the CTF (green star). Red rectangles indicate the major nesting sites in Grand Cayman and Little Cayman. **B** Distance between nests (Nest-site fidelity) of wild females laying more than 3 clutches per season ($n = 27$ between nest distances) based on geographic coordinates of nests. Red shows the mean distance between consecutive nests, while yellow shows the distance between the two most distant nests of the same season. Most females lay nests within a very short distance as can be appreciated in the pie chart in **C**, showing the percentage of wild females nesting within 1 km vs. more than 1 km.

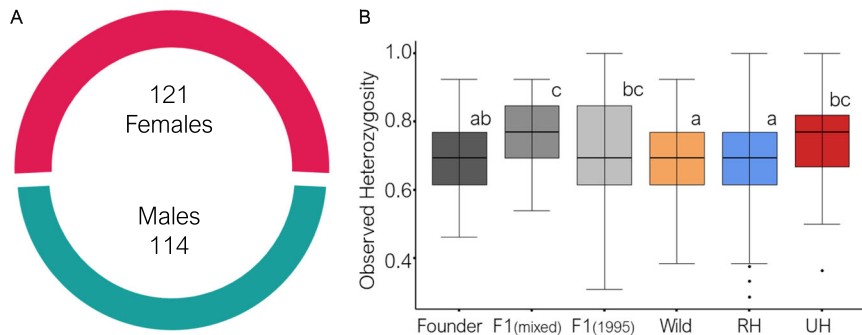

**Fig. 5 Analyses of biological parameters. A** Genetically estimated breeding population size of the Cayman Islands wild populations during the sampled period showing a sex ratio not significantly different from 1:1. Females include both wild individuals and females inferred by parentage analysis, while males were all inferred by parentage analysis. **B** Boxplots of observed heterozygosity values for the CTF breeding stock subgroups, wild females and nests related (RH) or unrelated (UH) to the CTF ($n = 634$ individuals). Boxplots sharing a lowercase letter showed non-significant differences according to two-sided Wilcoxon rank-sum tests, while boxplots not sharing any lowercase letter presented significant differences. The details of each pairwise Wilcoxon rank-sum can be found in the Supplementary Information, Table S5. The lower and upper hinges of each boxplot correspond to the first and third quartiles (the 25th and 75th percentiles). The upper whisker extends from the hinge to the largest value no further than 1.5 * IQR from the hinge (where IQR is the inter-quartile range). The lower whisker extends from the hinge to the smallest value at most 1.5 * IQR of the hinge. Data beyond the end of the whiskers are called 'outlying' points and are plotted individually. The line represents the median of all values.

a greater proportion of female offspring produced at temperatures above the pivotal temperature (~29 °C)[37,38] and a greater proportion of males produced at temperatures below this. In the Cayman Islands wild breeding adults, we did not detect female skewed sex ratios. The minimum genetically estimated breeding population size was of 121 females and 114 males (Fig. 5A), with

a sex ratio (male:female) of 0.94:1 not significantly different from a 1:1 proportion (d.f.=1, $\chi^2 = 0.053$, $p$ value = 0.817). Similar sex ratios were obtained by analysing each island independently (supplementary information: Table S2). This is somewhat at odds with studies on several Caribbean green turtle populations which suggested female-biased primary sex ratios[35] measured using

incubation temperatures. Nevertheless, adult sex ratios of the present study represent incubation conditions at least 15 years prior, either in the wild nesting beaches or in the CTF hatchery[27]. Consequently, future wild adults may be more female biased as a result of recent global warming.

To test the potential effect of the assisted colonisation on individuals' fitness (viability, fertilization success, clutch size and hatchling heterozygosity) we ran Linear Mixed-Effects Models with particular focus towards individual heterozygosity, and relatedness to the CTF. As described in previous studies[26,39], we found that larger females lay a significantly higher number of eggs per clutch (supplementary information: Table S3). Moreover, this study found that nests with larger clutches showed higher fertilization success (i.e. proportion of eggs that developed an embryo or hatchling), and nests with higher fertilization success showed higher viability (i.e. proportion of developed eggs hatched) (supplementary information: Table S3). Female and hatchling heterozygosity and CTF relatedness had no significant effect on the fertilization success or viability (supplementary information: Table S3), suggesting that individuals coming from the CTF program are not affecting the fitness of the new population. However, these results only refer to the first generation of wild hatchlings, due to the long generation time of the species[40,41]. Population fitness analyses should be repeated in the future to monitor potential drops in fitness due to outbreeding, since their deleterious effects can appear in later generations[42]. Our study sets the baseline to evaluate these components in future generations, but also provides the analytical framework to pursue these type of studies in other endangered species for which ex-situ conservation programs are envisaged.

When we compared heterozygosity values of hatchlings related and unrelated to the CTF, with the adult wild population and the subgroups of the CTF (i.e. Founders, F1(mixed) and F1(1995))[26] (Fig. 5B), related hatchlings had low heterozygosity levels as did the wild population and the founder group. On the other hand, unrelated hatchlings had higher levels of heterozygosity and similar values to all groups belonging to the CTF. A previous study showed that all CTF individuals have high levels of heterozygosity because founders were collected from genetically different populations, and first-generation breeders are descendants of founders and showed signs of outbreeding[26]. Wild hatchlings not related to the CTF could be the result of the mating of individuals coming from three different groups: i) individuals of external contribution (i.e. migrations from other populations); ii) individuals of the original wild population still nesting in the Cayman Islands; or iii) adult captive turtles escaped after Hurricane Michelle damaged the CTF facility in 2001. Unfortunately, these different hypotheses will remain untested, since genetic data from the original population and the initial CTF founder stock are not available.

**Assisted colonisation as a potential conservation measure**. As extinction risk projections predict an increasing number of species committed to extinction due to global warming[43], translocations and assisted colonisations seem a viable option for species facing challenging dispersal barriers[5]. Our results show that assisted colonisations as a conservation measure could possibly be used in sea turtles and might be adapted to other long-lived, migratory and philopatric species. Where habitat degradation undermines species survival[14], assisted colonisations might become an important future strategy to prevent extinctions for some species[5]. However, decision-making must include factors such as risk of decline or extinction under climate change, the technical possibility of the translocation and establishment of the species, and assessment of whether the translocation benefit

outweighs biological and socioeconomic constraints[14]. In this study, we show that assisted colonisation can establish sea turtle nesting populations and we also provide insight on factors related to feasibility and long-term viability. For sea turtles, important considerations for captive breeding include animal husbandry and welfare concerns, the potential for disease transfer through the release of animals from an intensive-rearing facility into the wild, high costs, and apparently low rates of recruitment of captive-bred individuals into wild nesting populations[25]. In this study, we showed the results from more than 25,000 yearling and hatchling turtles released in the 1980s[27]. This assists in elucidating the timescale and costs of establishing viable wild turtle nesting populations through assisted colonisation.

For all species, potential implications for the welfare of both captive and wild animals, dictate that thorough scientific assessments must underpin each step of assisted colonisation projects. For this reason, the study of the foundation of new populations using a multidisciplinary approach is crucial to improve assisted colonisations and to tailor conservation action plans to the target species. In fact, a careful study of the species and the colonisation area prior to the reintroduction is necessary to ensure the survival of translocated individuals but also to determine the potential impacts that the colonisation may have on the host ecosystem[5]. In the case of the Cayman Islands, this aspect was not considered and individuals from distant populations were incorporated to the breeding stock.

The scientific study and ongoing monitoring of conservation actions is just as important as their initial implementation and, in the case of the Cayman Islands, the impact of the captive breeding program on other Caribbean populations should be assessed in the future. The CTF reintroduction program raised concerns about its effectiveness, its suitability to recover the population and its economic costs. Despite controversies surrounding the Cayman Turtle Farm outcome, detailed analyses have shown how this attempt started almost 50 years ago was successful in contributing captive-raised individuals to the wild and that the first wild generations are fit to survive in their natural habitat. We show how the study of the foundation of new populations in vertebrates with complex life histories, such as marine turtles, can also provide relevant information on the generations required to establish a genetically differentiated new population or potential alterations of fitness. Assisted colonisation has shown potential in protecting a complex and highly migratory species in response to a critical population decline. However, ex-situ strategies should not replace, but aid in-situ conservation, and the latter should be considered as a conservation management priority before resorting to complicated, costly and controversial ex-situ conservation strategies.

## Methods
**Sampling and data collection**. This research complies with all relevant ethical regulations, including research and environmental regulations from the Cayman Islands Department of Environment and Cayman Islands Government. Sampling and data collection are regularly performed by the DOE to monitor nesting beaches of marine turtles on the islands of Little Cayman and Grand Cayman, in the Caribbean. Samples for this study were collected from 320 nests laid in Grand Cayman and Little Cayman (Fig. 1A) during 2013, 2014 and 2015 nesting seasons, from May to November (Fig. 1B, Fig. 2). This sampling effort corresponded to 58% of the 552 nests reported during these nesting seasons and locations. Since regular nesting was not recorded in these islands prior to 1998[25], these hatchlings can be considered to be the first generation after the foundation of the new wild population. For nests assigned to a sampled wild female (see results) we gathered information regarding nesting date and GPS location from the DOE database as well as the total number of hatched and unhatched eggs, following standard procedures[39], and the presence of an embryo in the unhatched eggs, as checked visually. Using these parameters, for each clutch we estimated the fertilization success, expressed as the proportion of the eggs that hatched or developed an embryo and the viability, expressed as the proportion of eggs hatched from those that hatched or developed an embryo (Supplementary Data 1). If the female was

present at the nest discovery, we recorded its identification by using metal flipper tags and Passive Integrated Transponders (PIT tags) and measured its Curved Carapace Length (CCL). All known female-offspring pairs ($n = 25$) were used as a control for genetic parentage identification. Samples were taken from the hatchling's margin of the carapace and up to three hatchlings per nest were sampled. Samples were obtained with a scalpel blade and stored in 100% ethanol.

**Laboratory analyses and genotyping.** The DNA of one hatchling sample per nest was extracted using the QIAamp Blood and Tissue Kit (Qiagen®) or using E.Z.N.A.® Tissue DNA kit (OMEGA Bio-tek), following the manufacturer's protocol. All samples were genotyped at 13 microsatellite loci (Supplementary Data 1), originally designed for different species of sea turtles that amplify and are polymorphic in green turtles[44] using the same protocols used in the previous studies[26]. In summary, microsatellite loci were amplified with two multiplex PCR sets with fluorescent dye labelled primers (6-FAM, HEX or NED). Each multiplex was amplified in a final volume of 5 µl, with 2.5 µl of Multiplex PCR Master Mix (Qiagen), 1.5 µl of primer mix and 1 µl of DNA. After amplification, 15 µl of ultrapure $H_2O$ Ecolab was added in each reaction tube and amplification success was assessed in an agarose gel. Microsatellite allele sizes were estimated in 2 µl of diluted amplified DNA, 0.5 µl of GeneScan 500 Liz Size standard (Applied Biosystems) and 12.5 µl of deionized formamide on an ABI 3730 DNA Analyzer (Applied Biosystems) at the Serveis Científico-Tècnics of the Universitat de Barcelona, and alleles were assigned using Genemapper v3.7. We sequenced 81 individuals for 800 bp of the D-Loop mitochondrial DNA using published protocols[26]. To summarize, the fragment was amplified in a reaction volume of 15 µl containing 5.08 µl of deionized water, 3 µl of PCR buffer 5× (GoTaq Promega), 1.8 µl of dNTPs (1 mm), 0.6 µl of MgCl2 (25 mm), 1.8 µl of bovine serum albumin, 0.3 µl of forward primer (10 µm), 0.3 µl of reverse primer (10 µm), 0.12 µl of GoTaq G2 Flexi DNA Polymerase (Promega, 5 U/µl), and 2 µl of DNA. The amplified DNA was purified with Exo-SAP (2 µl containing 0.4 U of EXO and 0.4 U of TSAP) using a single cycle of 37 °C for 15 min and 80 °C for 15 min. Then, 1 µl (5 µm) of the forward primer was added to the purified product (LCM15382) and dried at 80 °C for 30 min in order to be sequenced on an ABI 3730 automated DNA analyser (Applied Biosystems) at the Serveis Científico-Tècnics from Universitat de Barcelona. Haplotypes were assessed (Supplementary Data 1) using Bioedit v7.2.5[45] by comparison to the haplotype database maintained by the Archie Carr Center for Sea Turtle research (https://accstr.ufl.edu/).

We used GENALEX v6.503[46] to compute within group observed heterozygosity. Differences in observed heterozygosity between groups were assessed using Wilcoxon rank-sum tests as implemented in *R*.

For parentage and relatedness analyses, in addition to our hatchling samples, we used genetic data from a previous study[26] that included genotypes from 57 wild green turtle females nesting on Grand Cayman in 2013 and 2014, as well as 257 females breeding in the Cayman Turtle Farm (CTF). The CTF breeding stock dataset included original founders of the captive population ($n = 25$), a first-generation cohort of breeders born in 1995 corresponding to a single cohort breeders replacement strategy (F1(1995), $n = 189$, previously mentioned in the literature as C1995[26]) and a multicohort group of first-generation breeders corresponding to a continuous replacement strategy (F1(mixed), $n = 43$, previously mentioned in the literature as MCF1[26]) (Fig. 1B). Thus, considering new and published data, we recovered data from three to four generations of sea turtles (Fig. 2), including the breeding stock within the Cayman Turtle Farm (founder individuals and their F1 offspring), wild turtle nesting females (most of them farm-released females nesting in the wild that will be F1 or F2 offspring of the founder stock), and hatchlings from nests laid in the wild (offspring of the first generation of wild nesting females).

**Parentage analysis.** We performed parentage analysis using COLONY v2 software[47], which performs parentage assignment and reconstructs genotypes for unsampled parents, allowing the identification of family groups with sampled and unsampled females and males. We set the parameters to long run, high precision and error rate = 0.0001. All hatchlings were included in the analysis as offspring and the genotypes of 57 wild adult females from a previous study[26] were included as mothers. We used D-loop mtDNA information to build two exclusion files. The first one contained female-offspring pairs with different mtDNA haplotypes, as a hatchling cannot be the offspring of a female with a different haplotype. The second exclusion file contained sib-sib pairs with different mtDNA haplotypes, as two hatchlings cannot be mother siblings if they have different mtDNA haplotypes. These two exclusion files were included in the input of the program to refine the assignment. We checked the accuracy of COLONY by comparing the output with 25 female-offspring known pairs recorded during field observations (Supplementary Data 1). All matches were concordant between field observation and genetic assignment.

In addition to providing parentage and sibship relationships, the output of COLONY was used to infer the minimum genetic census of breeders based on the number of males and females identified or inferred by the program as parents of the analysed hatchlings.

In order to understand the impact of the CTF reintroduction program on the two populations we computed Queller and Goodnight relatedness estimator[48] using the program Coancestry v1.0.1.9[49] between the 320 hatchlings collected on

the two islands and the 257 CTF individuals genotyped in a previous study[26]. A pair of individuals was considered unrelated if its lower bound of 95% confidence interval was lower than 0.0001 and its $r$ value was less than or equal to 0.3069[50]. ML-Relate v1[51] was also used to estimate the relationship between individuals using a log-likelihood approach. We only accepted pairs of individuals found as related by both programs. As both ML-Relate and Coancestry programs returned several matches for each individual, it was not possible to establish the exact level of relationship between each clutch and the CTF. Therefore, we categorized individuals as either related or unrelated (Supplementary Data 2). Hatchlings of known relationship with the CTF (having a genotyped wild mother related to the CTF[26]) were included in these analyses as a control to assess the reliability of the programs. A total of 131 hatchlings were assigned a wild mother previously found to be related to the CTF. Of these, 120 hatchlings were confirmed as related to the CTF by the two relatedness programs, and 11 hatchlings were only confirmed by ML-Relate as half siblings of CTF individuals. We calculated the proportion of hatchlings related to the CTF for Little Cayman and Grand Cayman and tested for significant differences between islands with a Chi-squared test with Yates' continuity correction with R[52].

**Genetic differentiation.** To identify early signs of genetic structuring, we tested the level of genetic differentiation between the hatchlings sampled in Grand Cayman and Little Cayman and the CTF adult females using both nuclear and mitochondrial markers. As several nests were found to be laid by the same female (see Results), we used only one random nest per female laid during the same nesting season to avoid pseudoreplication. Using microsatellites, we calculated pairwise $F_{ST}$ and statistical significance through 999 permutations using GENALEX[46]. A previous study[26] found deviations of Hardy-Weinberg Equilibrium within the farm that are the result of the CTF breeding program (e.g. deficit of heterozygotes in the founders due to a Wahlund effect, as they came from distant populations, or excess of heterozygotes on the F1 resulting from the outcross of the founders from different origins). However, these deviations did not come from null alleles and thus are not expected to overestimate $F_{ST}$ values. In order to have additional support for genetic differentiation, we performed a Discriminant Analysis of Principal Components with the R package adegenet v2.1.5[53] using microsatellite markers, retaining 128 PCAs. The mitochondrial haplotype frequencies extracted from 74 independent nests (nests laid by different females as indicated by the parentage analysis) were also used to calculate pairwise $F_{ST}$ values between groups of samples, and significance (p-value ≤ 0.05) was assessed through an exact test using Arlequin v5.2[54]. A previous study using the same set of markers[55] showed that our mitochondrial marker has the statistical power to reliably detect differentiation on $F_{ST}$ values above 0.01 and our set of microsatellites can reliably detect differentiation on $F_{ST}$ values above 0.0025.

**Nesting fidelity and reproductive fitness.** With the results of the parentage analysis, we were able to link all the data collected in the field for each nest with an identified mother. We added to the dataset information on 16 female-offspring pairs recorded during night patrols but with no hatchling samples collected and genotyped (supplementary information: Table S4). We analysed intraseasonal nest-site fidelity (NSF) of wild females laying three or more nests in the same season ($n = 27$) using geographic coordinates of their nests within the same season. Distances between nests were obtained by measuring the coastline between the geographic coordinates of consecutive nests using Daft Logic (https://www.daftlogic.com/projects-google-maps-distance-calculator.htm#), an online tool to calculate distances with Google satellite maps. We used two different measures of NSF: the mean distance between consecutive nests and the maximum distance, that is the distance between the two most distant nests laid by the same female within a season. We also analysed interseasonal nest-site fidelity of 8 wild females found to lay nests in more than one nesting season. We used the same tool to calculate the geographic distance between nests laid by the same female in different seasons.

In order to assess any potential effect of females' heterozygosity on NSF we performed a generalised linear model in R[52] using the mean and the maximum distance between their nests as the response variable. We also performed a Kruskal-Wallis test to detect any impact of female relatedness to the CTF (subdivided in related, half siblings and unrelated, as found in Barbanti et al, 2019) on mean and maximum distance.

The parentage analysis performed by COLONY also provided the number of males and females that produced the offspring sampled on both Cayman Islands during the different nesting seasons. These data allowed us to estimate the minimum breeding population size across seasons. We added to the count three of the eight females recorded nesting during night patrols, which were not included in the genetic census because they had no nest assigned with genetics (supplementary information: Table S4). We calculated the sex ratio of the whole population and of each island separately, and performed a Chi-squared test with Yates' continuity correction[56] with R[52] to evaluate the significant difference in the number of males and females that would indicate skewed sex ratios.

We carried out Linear Mixed-Effects Models to detect possible impacts on nest fitness and hatchling observed heterozygosity caused by the reintroduction program as measured by female observed heterozygosity and relatedness with the CTF. We performed five different models for nest fitness using as response

variables clutch size, fertilization success (i.e. proportion of eggs with hatchlings or developed embryos), and viability of the clutch (i.e. proportion of eggs that hatched from fertilized eggs) (supplementary information: Table S3). Mother ID and the year of nesting season were set as random factors in all the models. We only considered data belonging to wild sampled females due to the lack of some parameters of the inferred females (curved carapace length, mother heterozygosity and mother relatedness to the CTF). We also evaluated the effect of nest laying date on fertilization success, viability, and clutch size. We considered the nesting date as the quartile of the nesting season in which the nest was laid, as nesting seasons can shift slightly in different years. We first calculated the duration of the nesting season as the period between the first and the last recorded nest of the season, and we then divided this period into quartiles to know in which quartile of the nesting season a particular nest was laid. The models were performed using the R package lme4 v1.1-28[57] and significance of categorical values were assessed with the package car v3.0-12[58]. Finally, we evaluated the levels of observed heterozygosity of hatchlings related and unrelated to the CTF, and we compared them with observed heterozygosity values of wild sampled females and CTFsubgroups[26] using Wilcoxon rank-sum tests as implemented in $R$[52].

**Reporting summary**. Further information on research design is available in the Nature Research Reporting Summary linked to this article.

## Data availability

All genotypes, parentage analyses detailed results and field data collected are included in the Supplementary Data 1 file. Sequences and Genebank Accession Numbers of the D-loop haplotypes are listed in Table S6. Source data are provided as a Source Data file. Source data are provided with this paper.

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

## Acknowledgements

This work was supported by the project CTM2017-88080 (MCIN/AEI/10.13039/501100011033 and by ERDF A way of making Europe of the European Union), the project PID2020-118550RB (MCIN/AEI/10.13039/501100011033), the Darwin Plus Project DPLUS019 Socio-economic aspects of turtle conservation in the Cayman Islands, funded through the Department for Environment, Food and Rural Affairs (Defra) in the UK and by the projects 9013 and 10084 of the Fundació Bosch I Gimpera-UB. C.C. and M.P. are part of the research group SGR2017-1120 of the Generalitat de Catalunya. A.B. was supported by grant 2017 FI_B 00997 of the Generalitat de Catalunya-AGAUR. We also thank Department of Environment staff members and volunteers, in particular Lucy Collyer and Joseph Roche Chaloner for their invaluable assistance with fieldwork coordination and sample collection.

## Author contributions

A.B., J.M.B, A.C.B., C.C., B.J.G. and M.P conceived and designed the study. J.M.B. led the Darwin project, which funded sample collection in Cayman Islands, together with A.P.V. who assisted with the coordination of fieldwork. A.C.B. and B.J.G. coordinated the application for permits and sample transportation. A.B., C.C. and M.P. designed the genetic analysis. A.B., M.T. and C.C. undertook the laboratory analysis. A.B., M.T., C.C. and M.P. conducted the data analysis with inputs from A.C.B., B.J.G., A.P.V. and J.M.B. A.B., M.P. and C.C. wrote the manuscript with input from all of the authors.

## Competing interests

The authors declare no competing interests.

## Additional information



