## [Peer Review File · Nature Communications]

Reviewers' Comments:

Reviewer #1:

Remarks to the Author:

This paper reports the success of an assisted colonization action taken for green turtles in the Cayman Islands. The translocations were undertaken 50 years ago, so a great deal can be learnt from the outcomes of that action. The dataset amassed and reported here is impressive, and the methods and analyses are appropriate and thorough. The results can help guide future conservation translocations. Overall, this is a comprehensive paper, which aggregates multiple lines of evidence. My primary comments relate to the presentation of the work.

1. My first comment is on the writing style in terms of structure. The main body text seems to move between rationale, methods, results, and context from sentence-to-sentence, so it is not always easy to tell what the objective of each part of the analysis is. I had to read parts of the text a couple of times to differentiate which inferences were based on the data versus the literature. I suggest the authors revise the main body text to clearly identify the goals/hypotheses of each part of the analysis, and delineate inferences from the data versus speculation/hypotheses based on the literature.

2. I recognize that this journal's format places Methods after the Discussion, however I do think the body text is a little Methods-heavy, which is probably a contributing factor to my first comment. I suggest the authors revise the main body text to reduce Methods detail (by moving material to the Methods section if not already present there), which would help improve the emphasis on key findings of the work, making those results easier for your reader to identify and interpret.

3. My final main comment is in regards to the innovation of the study. There are several parts to this paper, so it is not straightforward to discern what the primary lessons are. In part, this is a double-edged sword, because it seems that it is precisely the long-term, multidisciplinary nature of the study that is its key innovation. However, this was not clear to throughout the body text, until reading the final paragraphs of the paper. Clarifying the primary overarching goal/innovation early in the paper, and ensuring each component of the work is tied into that, will benefit the impact of this work.

Minor comments (some of which are examples of the above main comments):

- L16: the abstract opens with a focus on adaptation, but adaptation per se is not deeply explored in this paper (instead, the work focuses more on the population genetic ramifications of the translocation). Suggest reconsidering this framing.
- L88 (and elsewhere), "Here we present a study..." to...? The objective of this part of the analysis (as it relates to the main goals of the work) is not clear.
- L90 and following: this reads quite heavily as methods and could be abbreviated so that the results themselves are emphasized more clearly
- L109-110: this is an example of an unclear goal: what is being tested here?
- L136-137: this is another example where the objective of this part of the work (as related to the overall goals) could be stated more clearly/earlier
- L143-146: an example of heavy methods detail that could be abbreviated
- L156-159: I could not follow this sentence; possible editing error?
- L170-171: this claim seems to overstate the findings, which are based only on heterozygosity and relatedness. Many other elements of the reintroduction could impact nesting (e.g. behavioural change as a result of captivity)
- L199-202: it's not clear to me how outbreeding depression might be projected to worsen, when it is the founders that were maximally outbred. Is there something missing from this argument?
- L223: this inference would benefit from comparison to other species, to demonstrate the broader application of the findings
- L238-241: "We show...", I could not intuit which particular results support these claims (e.g. "complex vertebrate species", "evolutionary scales", etc). This statement may be an overgeneralization.
- L243-245: here, the implied main takehome of the paper seems only distantly connected to the

goal of the paper ("prevention of critical thresholds" is not analysed) – suggest clarifying the concluding statement in line with the overarching goal of the work.

- Figures: overall, the figures are elegantly presented. I found the various connecting lines of Figure 1C confusing though, particularly around the middle of the image where the lines intersect (is the order of intersection important?); did "population differentiation" information contribute to any of the inferences on the right?

Reviewer #2:

Remarks to the Author:

The manuscript by Barbanti et al. "The architecture of assisted colonisation in sea turtles" is in general an important contribution to the understanding of the success of reintroduction programs in sea turtles. The authors examine 634 green turtles (CTF female breeders; wild individuals) using D-loop and 13 microsatellite loci. In addition reproductive parameters have been collected from two wild populations. All in all, the paper is concise, generally well written, and provides adequate context to explain the value of the study. The raw data are provided (even though in the present form not really useful-see comments below) and given the methods in the manuscript the results should be repeatable. Nevertheless I have some reservations with some interpretations. See comments below:

Line 102-104: You classified your results from the relatedness analysis only into two classes – related or not. The program ML-Relate gives also the option to differentiate between PO=parent-offspring, FS= full sibling, Hs=half sibling... Could you please also include this information in the supplemental info? Since I cannot read the supplementary information properly, I am not sure if this information is available.

Line 109-117: I would be careful with the interpretation of this results. You are comparing hatchling populations of two islands with the populations of captive female breeders. I do believe that the sampling can bias your results due to Hardy-Weinberg presumption!

Line 117-120: It is true that the F_{st} values between the two islands are statistically significant, for nuclear microsatellites, but not for mtDNA. In addition, even though they are significant, you still need to compare your values to other wild living sea turtle populations to interpret the level of differentiation. F_{st} below 0.05 is considered in other turtle or tortoise populations as very low differentiation.

Line 122-132: As previously mention I would be careful by comparing this three populations and interpreting the results the way you do. Probably you need to rephrase this part and be more specific. Explain how you differentiate between the effects of genetic drift and philopatry using your data. It needs to be clear which results indicate that genetic drift is the driving force for structuring. How do you exclude selection as a force using microsatellite data? Please avoid to strong wording like "...genetic drift resultant from a founder effect is a strong evolutionary force, able..."

Line 131-132: I am not sure how you come to this conclusion "...generating significant genetic structuring in sea turtles in only one generation after the foundation of new populations." If the program started 50 years ago I would expect more than one generation? Please explain this better.

Line 181-182: Did you estimated the breeding population for both islands together? What happens if you estimate it separately, since you talk before about two genetically differentiated populations? In general, it would be good to see, if there is any link between genetic differentiation and ecological differentiation (sex ratio, nest fidelity...). This way you could support your hypothesis!

Line 203-211: Please add to Figure 4 an explanation on the letters (ab, a, bc...) which should show statistically significant groups based on observed heterozygosity according to Wilcoxon rank sum tests.

Figure 1: Are you sure the latitude values on the first bigger map are correct? Comparing the insert and the main map there is a big discrepancy in the latitude values.

Figure 2: Please include the % of the observed variation explained by the first five PCs.

Methods

Line 426: How many wild females have been sampled directly when laying the eggs? 16? Please include this information also in line 433.

Line 502: Please be more precise explaining the PCA analyses. Did you use only microsatellite data?

Supplementary Information was not really useful due to converting into the pdf. It was not possible to read it properly, so I can't comment on it!

Reviewer #3:

Remarks to the Author:

This manuscript addresses the potential for assisted colonization in marine turtle conservation using the Cayman Island green turtle nesting populations as a case study. It builds on previous research demonstrating close relationships of Little Cayman and Grand Cayman nesting females to the captive Cayman Turtle Center breeding population, and yet also differentiation of all three, through additional years of data from the wild nesting population. Novel analyses of female nest site fidelity and hatchling viability suggest that neither have been negatively impacted by the reintroduction program. This paper will be of interest to marine turtle biologists who can use these approaches to address recent natural colonization as well as those contemplating management intervention to establish new assurance populations. It should be of broad interest to conservation biologists.

The methodology is clearly described and sound. I found the statistical approaches appropriate for addressing the questions raised.

Some questions

You note in the methods that you used mitochondrial DNA to exclude illogical mother-offspring and maternal sibling pairs inferred from the microsatellites. How many of these were there based on relatedness analyses from both programs? I didn't catch that in the methods or results, but it would be good to report this to help demonstrate the relative power of the microsatellite panel beyond the other controls (known mother-offspring and known CTF related wild nest mothers) implemented in the study.

It is interesting that such a large percentage of wild nests can be linked to a CTF breeder via relatedness, even given the sampling constraints identified. I appreciate that any wild nesters/nests that are not related to sampled CTF breeders may represent the remnant original population, new wild colonizers, or CTF individuals that died or escaped prior to the genetic sampling effort. However, for any nests that were related to CTF, would it be possible/interesting to assess the proportion of nests where BOTH parents appear related to CTF stock?

Minor editorial suggestions:

Line 101: "were" deceased?

Line 468: refine the "assignment":

Line 470: "maternal" siblings

Line 532: "These" data allowed

REVIEWER COMMENTS

Reviewer #1

1. My first comment is on the writing style in terms of structure. The main body text seems to move between rationale, methods, results, and context from sentence-to-sentence, so it is not always easy to tell what the objective of each part of the analysis is. I had to read parts of the text a couple of times to differentiate which inferences were based on the data versus the literature. I suggest the authors revise the main body text to clearly identify the goals/hypotheses of each part of the analysis, and delineate inferences from the data versus speculation/hypotheses based on the literature.

A: Following the reviewer's suggestion, we have clarified the goals and aims of each analysis in the main text within each section (lines 81-85, 92-97, 155-159, 334-335) as well as specified where our data is mentioned instead of literature.

2. I recognize that this journal's format places Methods after the Discussion, however I do think the body text is a little Methods-heavy, which is probably a contributing factor to my first comment. I suggest the authors revise the main body text to reduce Methods detail (by moving material to the Methods section if not already present there), which would help improve the emphasis on key findings of the work, making those results easier for your reader to identify and interpret.

A: We thank the reviewer for the comment. As suggested, we reduced the amount of methodology explained in the main text to allow more space for results and related discussion and moved these sections to methods when appropriate (e.g. lines 455-464, 493-498, 505-511).

3. My final main comment is in regards to the innovation of the study. There are several parts to this paper, so it is not straightforward to discern what the primary lessons are. In part, this is a double-edged sword, because it seems that it is precisely the long-term, multidisciplinary nature of the study that is its key innovation. However, this was not clear to throughout the body text, until reading the final paragraphs of the paper. Clarifying the primary overarching goal/innovation early in the paper, and ensuring each component of the work is tied into that, will benefit the impact of this work.

A: Following the reviewer's suggestion, we have clarified the main innovation of our study at the beginning (lines 81-85) and throughout the manuscript, hoping the major innovation point can now be understood and followed throughout.

Minor comments (some of which are examples of the above main comments):

- L16: the abstract opens with a focus on adaptation, but adaptation per se is not deeply explored in this paper (instead, the work focuses more on the population genetic ramifications of the translocation). Suggest reconsidering this framing.

A: Assisted colonisation is presented as an alternative conservation strategy to aid survival of species (adaptation) in the current changing conditions. However, we agree with the comment of the reviewer that starting the Summary mentioning 'adaptation' can add some

confusion. Thus, and following the reviewer's suggestion, we have rephrased the first sentence of the summary (line 16).

- L88 (and elsewhere), "Here we present a study..." to...? The objective of this part of the analysis (as it relates to the main goals of the work) is not clear.

A: As suggested by the reviewer (and also linked with reviewer's main comment 3), we rephrased this sentence to highlight the main goals and innovation points of the study (lines 92-97).

- L90 and following: this reads quite heavily as methods and could be abbreviated so that the results themselves are emphasized more clearly

A: As suggested by the reviewer, we removed part of the methodology explanation to leave more space to the results and their discussion (lines 96-98 with parts moved to lines 493-498, also with some methods expanded as requested by other reviewers 505-511 or 525-538).

- L109-110: this is an example of an unclear goal: what is being tested here?

A: We have rephrased this sentence to clarify what is tested here: the differentiation of the new populations (lines 113-118).

- L136-137: this is another example where the objective of this part of the work (as related to the overall goals) could be stated more clearly/earlier

A: We also have clarified the goals of this part related to the overall goal of the paper (lines 155-159).

- L143-146: an example of heavy methods detail that could be abbreviated

A: As suggested by the reviewer, we reduced the methodological explanations to leave more space to the results and their discussion (157-159).

- L156-159: I could not follow this sentence; possible editing error?

A: We rephrased the sentence for clarity (lines 168-171).

- L170-171: this claim seems to overstate the findings, which are based only on heterozygosity and relatedness. Many other elements of the reintroduction could impact nesting (e.g. behavioural change as a result of captivity)

A: As suggested by the reviewer, we have rephrased the sentence to lower the tone of our conclusions (lines 182-184).

- L199-202: it's not clear to me how outbreeding depression might be projected to worsen, when it is the founders that were maximally outbred. Is there something missing from this argument?

A: Although the founder captive population results to be outbred as a consequence of the individuals' origin, we have not detected any sign of outbreeding depression, nor does the

new wild population, at the moment. However, although we have not detected any signs of outbreeding depression, this process could manifest itself in the long term as shown in previous studies for other species of conservation concern. For this reason, we stressed the importance of future monitoring of this population. In order to clarify this point, we have modified this sentence (lines 213-215).

- L223: this inference would benefit from comparison to other species, to demonstrate the broader application of the findings

A: We rephrased this part (lines 233-237). Furthermore, our reference number 3, cited in this part, is a good review of assisted colonisations and conclude that this measure can have a broader application in several species.

- L238-241: “We show...”, I could not intuit which particular results support these claims (e.g. “complex vertebrate species”, “evolutionary scales”, etc). This statement may be an overgeneralization.

A: We have rephrased his sentence to clarify that our research can be used as a case study for other marine vertebrates with complex life histories rather than overgeneralize our findings. (lines 243-251).

- L243-245: here, the implied main takehome of the paper seems only distantly connected to the goal of the paper (“prevention of critical thresholds” is not analysed) – suggest clarifying the concluding statement in line with the overarching goal of the work.

A: We appreciate the reviewer’s comment. The last message of our study is aimed to underline the fact that, although assisted colonisations might be a solution to the extinction of some population or species, it should not be performed without scientific assessment or without considering other alternatives less controversial and less costly. Consequently, assisted colonisations should not be considered the first resort and in-situ conservation should not be replaced by ex-situ strategies unless necessary. As mentioned above we authors feel it is important to maintain a balanced focus in this study regarding assisted colonisation. To emphasize and clarify this message, specially linked to the overall aim of the study, we have rephrased this whole section (lines 233-274).

- Figures: overall, the figures are elegantly presented. I found the various connecting lines of Figure 1C confusing though, particularly around the middle of the image where the lines intersect (is the order of intersection important?); did “population differentiation” information contribute to any of the inferences on the right?

A: The lines represent how the different sources of data and outputs of the analyses interact to provide the different results. As an example, parentage assignments and CTF relatedness interact with each other and with reproductive data to provide inferences of differential fitness. As the reviewer guessed, the “population differentiation” information is an analysis on its own that do not contribute to the inferences on the elements in the right of the figure. We have added some explanation in the figure caption to explain the meaning of the connecting lines (lines 398-401).

Reviewer #2 (Remarks to the Author):

Line 102-104: You classified your results from the relatedness analysis only into two classes – related or not. The program ML-Relate gives also the option to differentiate between PO=parent-offspring, FS= full sibling, Hs=half sibling... Could you please also include this information in the supplemental info? Since I cannot read the supplementary information properly, I am not sure if this information is available.

A: Since the program MLrelate runs an all-against-all analysis, most clutches were found related to more than one individual from the CTF and with more than one relationship level (PO, FS, HS). For this reason, we used more than one program (coancestry) so to refine this analysis. Consequently, it is unfeasible to provide all the potential levels of relationship with all individuals in the supplementary table. Instead, we have summarised the data in a table (Table S4) where each row represents a clutch and each column represents a relatedness level (PO, FS, HS and coancestry) to CTF individuals. The data in each cell indicates how many females from the CTF are related to each clutch. We hope this kind of solution is satisfactory as supplementary information in addition to the table already provided in the previous version. This is also clarified in the Methods (lines 525-528)

Line 109-117: I would be careful with the interpretation of this results. You are comparing hatchling populations of two islands with the populations of captive female breeders. I do believe that the sampling can bias your results due to Hardy-Weinberg presumption!

A: As the reviewer points, our previous paper (reference 3, Barbanti et al 2019) already found deviations from HWE within the farm and in the wild individuals. These results are expected from the foundation and maintenance of a captive population and posterior reintroduction (e.g. excess of heterozygotes on the F1 resulting from the outcross of the founders from different origins). Deviations from HWE do not come from null alleles that could overestimate the F_{ST} values (we checked for null alleles with MICRO-CHECKER as in the above-mentioned paper), but on the expected genetic processes on a captive breeding reintroduction program. Furthermore, the DAPC (which plots each individual independently, Figure 2C) indicates that the considered groups are differentiated. Considering all the evidence, we can assume with confidence that the genetic differentiation found is reliable. In order to explain these details without breaking the flow of the main text, we have included them in the methods section (lines 545-550).

Line 117-120: It is true that the F_{ST} values between the two islands are statistically significant, for nuclear microsatellites, but not for mtDNA. In addition, even though they are significant, you still need to compare your values to other wild living sea turtle populations to interpret the level of differentiation. F_{ST} below 0.05 is considered in other turtle or tortoise populations as very low differentiation.

A: Unfortunately, the comparison with other nesting populations for nuclear DNA is not possible, as the genotyping needs to be undertaken using the same set of microsatellites and calibration among laboratories is needed. However, we have added a reference from another study using the same set of microsatellites in the same species to show the robustness of our set of microsatellites. Particularly, Bradshaw et al 2018, using the same set of microsatellites found significant differentiation within Cypriot nesting populations at F_{ST} values around 0.005 (a lower value than ours) and the authors tested the statistical power of these set of markers showing that they were powerful enough to detect differentiation on F_{ST} values above 0.0025. We have added a few lines on the methods explaining the threshold of

F_{ST} values for which our set of markers are reliable to detect significant genetic differentiation (lines 556-559).

Line 122-132: As previously mention I would be careful by comparing this three populations and interpreting the results the way you do. Probably you need to rephrase this part and be more specific. Explain how you differentiate between the effects of genetic drift and philopatry using your data. It needs to be clear which results indicate that genetic drift is the driving force for structuring. How do you exclude selection as a force using microsatellite data? Please avoid to strong wording like "...genetic drift resultant from a founder effect is a strong evolutionary force, able..."

A: Considering the genetic differentiation found in only very few generations, the two possible explanations are a strong selection or genetic drift due to the potential low number of individuals that originated the population. The fact that the two populations diverged in different directions (Figure 2C) and that we found no evidence of fitness effects associated to the influence of the CTF suggest that a strong selection is not very likely, while a strong genetic drift seems a more plausible explanation causing the divergence found. Philopatry would help maintaining the populations isolated in future generations and thus increase the differentiation. We have revised this part of the text to clarify these concepts (lines 131-144).

Line 131-132: I am not sure how you come to this conclusion "...generating significant genetic structuring in sea turtles in only one generation after the foundation of new populations." If the program started 50 years ago I would expect more that one generation? Please explain this better.

A: We appreciate the reviewer's comment. There are two sources of evidence to support this statement and the fact that we can only have up to F3 individuals in our sampling (compared to the founders). On one hand, the CTF breeding population is composed exclusively by the founder individuals and their offspring (F1 generation), meaning that initial wild nesting females related to the CTF are going to be F1 or F2 individuals. On the other hand, sea turtles have very long generation times and although the program started 50 years ago, the individuals released began nesting in the Cayman Islands in 1998 (we have added a recent publication to illustrate this and replace former Figure S1); virtually no nests had been detected in the Islands for years prior to this year. Based on the time that sea turtles need to reach maturity in the wild (20-25 years) it would be very unlikely that individuals born between 1998 and 2012 in the wild would have reached maturity and laid nests before our sampling period 2013-2015. This means that our sampled hatchlings can be the F2 or F3 at maximum, and we can only talk about one generation of wild sea turtles after foundation. To avoid filling the main text with this explanation, we have added a general statement in the main text (lines 92-94) and a detailed information in the Methods (lines 493-498).

Line 181-182: Did you estimated the breeding population for both islands together? What happens if you estimate it separately, since you talk before about two genetically differentiated populations? In general, it would be good to see, if there is any link between genetic differentiation and ecological differentiation (sex ratio, nest fidelity...). This way you could support your hypothesis!

A: We thank the reviewer for the comment. We did calculate the breeding population for the two islands together and separately, and also computed the sex ratio for the two populations

separately as shown in supplementary table 2. Sex ratio were not significantly different between sexes nor between the two islands. This is also explained in the text at lines 192-196.

Line 203-211: Please add to Figure 4 an explanation on the letters (ab, a, bc...) which should show statistically significant groups based on observed heterozygosity according to Wilcoxon rank sum tests.

A: We have extended the explanation to the letters of Figure 4B for clarity (439-441). Boxplots sharing a lowercase letter showed non-significant differences according to Wilcoxon rank sum tests, while boxplots not sharing any lowercase letter presented significant differences.

Figure 1: Are you sure the latitude values on the first bigger map are correct? Comparing the insert and the main map there is a big discrepancy in the latitude values.

A: The map has been corrected as indicated.

Figure 2: Please include the % of the observed variation explained by the first five PCs.
Methods

A: The information about the variation explained by the PCs has been incorporated as an inset in Figure 2C and explained in the legend (lines 414-415).

Line 426: How many wild females have been sampled directly when laying the eggs? 16?
Please include this information also in line 433.

A: This information has been included (line 467).

Line 502: Please be more precise explaining the PCA analyses. Did you use only microsatellite data?

A: We have detailed that we used only microsatellite markers (line 544).

Reviewer #3 (Remarks to the Author):

You note in the methods that you used mitochondrial DNA to exclude illogical mother-offspring and maternal sibling pairs inferred from the microsatellites. How many of these were there based on relatedness analyses from both programs? I didn't catch that in the methods or results, but it would be good to report this to help demonstrate the relative power of the microsatellite panel beyond the other controls (known mother-offspring and known CTF related wild nest mothers) implemented in the study.

A: We thank the reviewer for this comment as we see that this needs further explanation. The mtDNA haplotypes that we obtained were used to build two types of exclusion files, one with a list of mothers-offspring pairs that could not be possible (if having different mtDNA haplotype) and another with a list of offspring pairs that could not be mother half sibs (if having different mtDNA haplotypes). These two files were introduced in COLONY (the only program that does direct assignments) from the beginning to aid parentage analysis. We have rephrased this section to clarify how this exclusion is done (lines 505-511).

It is interesting that such a large percentage of wild nests can be linked to a CTF breeder via relatedness, even given the sampling constraints identified. I appreciate that any wild nesters/nests that are not related to sampled CTF breeders may represent the remnant original population, new wild colonizers, or CTF individuals that died or escaped prior to the genetic sampling effort. However, for any nests that were related to CTF, would it possible/interesting to assess the proportion of nests where BOTH parents appear related to CTF stock?

A: We identify two possible cases here, linking an individual (a nesting female) to the CTF and linking the parents of an individual (a hatchling) to the CTF. In the first case, though that kind of analysis would be interesting, unfortunately we do not have at our disposal samples of male breeders from the CTF and therefore, it would not be possible to assess if both parents of wild females could belong to the CTF. In the second case, all the males of the hatchlings, and some of the females, are inferred by the program, meaning that we don't have their real genotypes to compare with the individuals of the farm. Thus, although the analysis suggested by the reviewer would be very interesting, it is not possible with our current dataset.

Minor editorial suggestions:

Line 101: "were" deceased?

A: Changed as suggested.

Line 468: refine the "assignment":

A: Changed as suggested.

Line 470: "maternal" siblings

A: This section has already changed due to a previous comment of the reviewer.

Line 532: "These" data allowed

A: Changed as suggested.

Reviewers' Comments:

Reviewer #1:

Remarks to the Author:

This manuscript, which I reviewed previously, uses genetic pedigree and relatedness reconstruction to evaluate the outcome of an assisted colonization of sea turtles. The authors have done a good job of responding to most of my main comments.

One of my main comments in my previous review related to the novelty of the research for publication in a general biology journal such as Nature Communications. In that review, I considered that the study's long-term dataset was its most innovative component. The revised paper is much clearer, but I can now see that although the research studies data from three to four generations (L92), the fitness consequences of the translocation – arguably the most important part – are only reported for the first WILD generation (L213). As the authors point out – and I agree – fitness problems can manifest in the 2nd and later generations. This dataset therefore only provides partial information toward the effects of the assisted colonization. The authors reiterate the long-term nature of their dataset, but the connection between that history and its merit for determining the effect of assisted colonization is still unclear to me, being that only one generation in the wild was studied. Perhaps a schematic illustrating the populations' history and key events, alongside predictions in relation to the study design, would help.

Minor comments:

L133, 139-141 – it's not clear what statistics enable the authors to conclude that this pattern is a result of "genetic drift during founder process" (= "founder effect"?) as opposed to selection, because selection was not tested. How can they be sure it's drift? One way to test it would be to use with a randomization model to determine whether the observed results are in line with simulated neutral (chance) processes.

L152 – what is "evolutionary behavior" (is it simply "behavior"?)

L179, 181, 195 and elsewhere – to report their statistics in full, the authors should include degrees of freedom or sample sizes with each (including t, Chi-squared, and other tests).

L193 – if I am reading correctly, then this is the sex ratio of the breeders, not the sex ratio of the population. It is still possible that there could be a skew in the population as a whole, or in the first-generation offspring produced. Were sex-linked genetic markers available to test this?

L203 – reiterate briefly here how "fitness" was defined

L257 – authors state that careful assessment of recipient ecosystem is required; was this done?

How did the outcome affect the method/results?

L263, 273 – what controversies are the authors alluding to?

L268 – I did not see an analysis of "evolutionary time". How was this shown?

The figures were reproduced with quite low resolution on the version I received – suggest checking Table S1 – give the sample size of each site

Table S3 – full regression statistics are not reported, only p-values. Authors also need to include effect sizes and errors for all parameters, including the intercept. They should also report model fit statistics (e.g. R-squared) and the error structure used for each model.

Reviewer #2:

Remarks to the Author:

The authors followed my suggestions and answered in detail my concerns. I have no more comments.

Reviewer #3:

Remarks to the Author:

I am satisfied with the revised version of the manuscript. The revisions have clarified the ambiguities with respect to methodology and interpretations based on these approaches. I concur with the authors on the interpretation of significant nuclear differentiation despite the low FST value. I also agree that it's difficult to generalize across studies that use different loci given

variation in polymorphism among them that place a ceiling on F_{ST} values. Addition of the Bradshaw et al. reference is helpful for context on this front.

Find below how we answered the comments of Reviewer #1 (as Reviewers #2 and #3 had no additional requests) . We included the original comments of the reviewer and we have preceded our answers by ‘*A:’ and highlighted in blue for clarity.

REVIEWER COMMENTS

Reviewer #1 (Remarks to the Author):

This manuscript, which I reviewed previously, uses genetic pedigree and relatedness reconstruction to evaluate the outcome of an assisted colonization of sea turtles. The authors have done a good job of responding to most of my main comments.

One of my main comments in my previous review related to the novelty of the research for publication in a general biology journal such as Nature Communications. In that review, I considered that the study’s long-term dataset was its most innovative component.

The revised paper is much clearer, but I can now see that although the research studies data from three to four generations (L92), the fitness consequences of the translocation – arguably the most important part – are only reported for the first WILD generation (L213). As the authors point out – and I agree – fitness problems can manifest in the 2nd and later generations. This dataset therefore only provides partial information toward the effects of the assisted colonization. The authors reiterate the long-term nature of their dataset, but the connection between that history and its merit for determining the effect of assisted colonization is still unclear to me, being that only one generation in the wild was studied.

A: Green turtles are long lived animals and their age at sexual maturity estimates are highly variable (8-12 years in captivity and 18-44 years in the wild) due to feeding, origin and methodology used (Goshe *et al.* 2010, Bjorndal *et al.* 2013). The time since the recovery of the Cayman Islands beaches with a few green turtles wild nests (Figure 2), along with high juvenile mortality and long lifespan allows to study the relatedness and fitness components only for the first wild generation. Importantly, our study sets the baseline to evaluate these components in future generations but also provides the analytical framework to pursuit this type of studies in other endangered species for which ex-situ conservation programs are envisaged. We have included this information and references in the manuscript for clarity (lines 217-2018 and lines 220-222).

- Goshe, L.R., Avens, L., Scharf, F.S. et al. Estimation of age at maturation and growth of Atlantic green turtles (*Chelonia mydas*) using skeletochronology. *Mar Biol* 157, 1725–1740 (2010). <https://doi-org.sire.ub.edu/10.1007/s00227-010-1446-0>
- Bjorndal, K.A., Parsons, J., Mustin, W. et al. Threshold to maturity in a long-lived reptile: interactions of age, size, and growth. *Mar Biol* 160, 607–616 (2013). <https://doi-org.sire.ub.edu/10.1007/s00227-012-2116-1>

Perhaps a schematic illustrating the populations’ history and key events, alongside predictions in relation to the study design, would help.

A: Following the reviewer suggestion, we have included a figure (new Figure 2) that incorporates a timeline of the Cayman Island population History and the reintroduction

program including the key events and processes addressed in this study. With this new figure we aim to clarify any doubt as well as to give an overall view of this reintroduction program.

Minor comments:

L133, 139-141 – it's not clear what statistics enable the authors to conclude that this pattern is a result of "genetic drift during founder process" (= "founder effect"?) as opposed to selection, because selection was not tested. How can they be sure it's drift? One way to test it would be to use with a randomization model to determine whether the observed results are in line with simulated neutral (chance) processes.

A: The two Cayman Islands studied are only separated 108 Km, have similar environmental characteristics and have differentiated in only one to few generations. Considering all these evidences, we suggest that genetic drift (founder effect) is driving the differentiation between them, as selection coefficients would need to be strongly different among the tested groups to generate this differentiation in this few generations. We have added a sentence to stress this point "Selection is unlikely to have played a major role in this differentiation process, as the two islands are separated by only 108 km and have similar environmental conditions. Only strong and opposite selection coefficients could explain such differentiation in a few generations" (lines 135-138).

L152 – what is "evolutionary behavior" (is it simply "behavior"?)

A: We indicate evolutionary behaviour since philopatry is observed in all sea turtles. To clarify this point we have change it to "this successful behaviour that is common in all extant turtles" (line 155-156).

L179, 181, 195 and elsewhere – to report their statistics in full, the authors should include degrees of freedom or sample sizes with each (including t, Chi-squared, and other tests).

A: We have added the requested information.

L193 – if I am reading correctly, then this is the sex ratio of the breeders, not the sex ratio of the population. It is still possible that there could be a skew in the population as a whole, or in the first-generation offspring produced. Were sex-linked genetic markers available to test this?

A: We indicate that it is the minimum number of breeders of the population in Cayman Islands considering all years and islands analysed. As indicated in former line 189 (present line 192) sea turtles exhibit Temperature-dependent Sex Determination. Furthermore, in our previous version of the manuscript we already discussed potential changes in sex ratios (former lines 198-201, present lines 201-204).

L203 – reiterate briefly here how "fitness" was defined

A: As suggested we have described in brackets which fitness components (viability, fertilization success, clutch size and hatchling heterozygosity) are being analysed (as detailed in the Material and Methods). Lines 206-207.

L257 – authors state that careful assessment of recipient ecosystem is required; was this done? How did the outcome affect the method/results?

A: This was not done in the present reintroduction program although we believe it is important to be considered in future similar programs. For instance, founders of the captive population were translocated from different genetic groups, others that present in the area. This should be avoided since it can impact natural populations of the whole area and/or affect the reintroduced individuals. It is thus very important not only to assess the fitness effects on this first generation of wild individuals but in future generations. This is affecting all species where translocation plants are being envisaged. We have rewritten this part to clarify this reasoning (lines 263-268).

L263, 273 – what controversies are the authors alluding to?

A: During the last years there have been different groups opposing to the Cayman Turtle Farm. The reasons are multiple ranging from the suitability to the reintroduction program to recover the species, its effectiveness or its economic costs. We have included this information for clarity (lines 272-273).

L268 – I did not see an analysis of “evolutionary time”. How was this shown?

A: We show that with few generations genetic differentiation is found probably to founder effect. We have used “generations” instead to avoid confusion (line 279).

The figures were reproduced with quite low resolution on the version I received – suggest checking

A: We have included the figures with a higher quality and provided also as separated files.

Table S1 – give the sample size of each site

A: The sample size has been included in the figure legend. Grand Cayman (n=115), Little Cayman (n=34) and Cayman Turtle Farm (n=257)

Table S3 – full regression statistics are not reported, only p-values. Authors also need to include effect sizes and errors for all parameters, including the intercept. They should also report model fit statistics (e.g. R-squared) and the error structure used for each model.

A: The information of each model has been included in Table S3